# Multiple fermion scattering in the weakly coupled spin-chain compound YbAlO$_3$

S. E. Nikitin [1,2,12 ✉], S. Nishimoto [3,4], Y. Fan [5,13], J. Wu [6], L. S. Wu [7,8], A. S. Sukhanov [1,2], M. Brando [1], N. S. Pavlovskii [9], J. Xu [10,14], L. Vasylechko [11], R. Yu [5 ✉] & A. Podlesnyak [7]

The Heisenberg antiferromagnetic spin-1/2 chain, originally introduced almost a century ago, is one of the best studied models in quantum mechanics due to its exact solution, but nevertheless it continues to present new discoveries. Its low-energy physics is described by the Tomonaga-Luttinger liquid of spinless fermions, similar to the conduction electrons in one-dimensional metals. In this work we investigate the Heisenberg spin-chain compound YbAlO$_3$ and show that the weak interchain coupling causes Umklapp scattering between the left- and right-moving fermions and stabilizes an incommensurate spin-density wave order at $\mathbf{q} = 2\mathbf{k}_F$ under finite magnetic fields. These Umklapp processes open a route to multiple coherent scattering of fermions, which results in the formation of satellites at integer multiples of the incommensurate fundamental wavevector $\mathbf{Q} = n\mathbf{q}$. Our work provides surprising and profound insight into bandstructure control for emergent fermions in quantum materials, and shows how neutron diffraction can be applied to investigate the phenomenon of coherent multiple scattering in metals through the proxy of quantum magnetic systems.

[1] Max Planck Institute for Chemical Physics of Solids, Dresden, Germany. [2] Institut für Festkörper- und Materialphysik, Technische Universität Dresden, Dresden, Germany. [3] Department of Physics, Technical University Dresden, Dresden, Germany. [4] Institute for Theoretical Solid State Physics, IFW Dresden, Dresden, Germany. [5] Department of Physics and Beijing Key Laboratory of Opto-Electronic Functional Materials and Micro-Nano Devices, Renmin University of China, Beijing, China. [6] Tsung-Dao Lee Institute and School of Physics and Astronomy, Shanghai Jiao Tong University, Shanghai, China. [7] Neutron Scattering Division, Oak Ridge National Laboratory, Oak Ridge, TN, USA. [8] Department of Physics, Southern University of Science and Technology, Shenzhen, China. [9] Kirensky Institute of Physics, Federal Research Center, Krasnoyarsk, Russia. [10] Helmholtz-Zentrum Berlin für Materialien und Energie, Berlin, Germany. [11] Lviv Polytechnic National University, Lviv, Ukraine. [12] Present address: Paul Scherrer Institute, Villigen PSI, Villigen, Switzerland. [13] Present address: Beijing National Laboratory for Condensed Matter Physics and Institute of Physics, Chinese Academy of Sciences, Beijing, China. [14] Present address: Heinz Maier-Leibnitz Zentrum, Technische Universität München, Garching, Germany. ✉email: stanislav.nikitin@psi.ch; rong.yu@ruc.edu.cn

The one-dimensional (1D) antiferromagnetic (AFM) XXZ spin $S = 1/2$ chain in a magnetic field (Eq. (1)) is among the most attractive, simple and, therefore, well-studied models in solid-state physics, from both experimental[1–4] and theoretical[5–7] points of view. Its Hamiltonian reads

$$\mathcal{H} = J \sum_i \left( S_i^x S_{i+1}^x + S_i^y S_{i+1}^y + \Delta S_i^z S_{i+1}^z \right) - \sum_i H_z S_i^z, \quad (1)$$

where $\Delta$ parameterizes the exchange anisotropy and $H_z$ refers to the applied longitudinal magnetic field. Depending on $H_z$, the XXZ model exhibits three types of low-lying excitations: (i) fractionalized fermionic spinons at zero field; (ii) bosonic magnons in the field-polarized regime; (iii) complex quasiparticles known as psinons and psinon-antipsinons at intermediate field, which adiabatically connect spinons and magnons, respectively[8]. Note that though the psinons are not canonical fermions, their low-energy dispersion can be linearized, thus the low-energy excitations can be considered as right and left-moving fermions. These fermions are fractionalized quasiparticles, which means that any local operator (such as a spin operator, responsible for neutron-sample interaction in the inelastic neutron scattering (INS) cross-section) creates more than one excitation, and therefore their single-particle bandstructure is not directly accessible by spectroscopic techniques. However, the multi-particle excitation spectrum manifests itself in the dynamical spin susceptibility, $\chi''(\mathbf{q}, \omega)$, which can be probed by INS.

Having a system with a gapless spectrum of fermionic quasiparticles, one can predict its physical properties using approaches developed for conventional metals. In particular, the realization of the Fermi surface (FS)—the most important concept in the

physics of metals—was discussed widely in the context of quantum spin liquids[9–12] and Heisenberg or XY spin chains, and in the latter case, the FS is reduced to two points at $\pm k_F$, as represented in Fig. 1a–c. Compared with electrons in a metal, the fermions in a spin chain have a significant advantage for experimentalists—by applying a magnetic field one can control the chemical potential directly without the doping-induced disorder, the need to perform complicated gating, and other problems intrinsic to electronic systems. The magnetic field effectively shifts the chemical potential of the fermion band and the nesting wavevector connecting the two branches follows directly the field-induced magnetization, $2k_F = \pi(1 \pm 2m)$ (where $m$ is the magnetization normalized by its saturation value, $M/M_S$)[13]. In the Ising regime ($\Delta > 1$), the $2k_F$ scattering gives rise to a gapless incommensurate longitudinal mode at finite fields[14]. This mechanism is similar to FS nesting and stabilizes an incommensurate spin-density wave (SDW) order at $q = 2k_F$ when the interchain coupling is present. Even though theoretical predictions for this SDW ordering have existed for a long time, its experimental realization is limited to very few materials including $BaCo_2V_2O_8$ and $SrCo_2V_2O_8$[15,16], both having a moderate Ising anisotropy, $\Delta \simeq 2$.

Recently, field-induced SDW order was observed in the Heisenberg spin-chain material $YbAlO_3$[17]. In this compound, the combination of strong spin-orbit coupling (SOC) and crystalline electric field lifts the degeneracy of the ground-state multiplet and causes an effective $S = 1/2$ at each Yb site at low temperatures. Despite the strong SOC of the Yb ion, the dominating intrachain exchange interaction $J = 0.21$ meV is almost isotropic, with $\Delta \simeq 1$[17,18] and the excitation spectrum above $T_N$ exhibits spinon continuum as shown

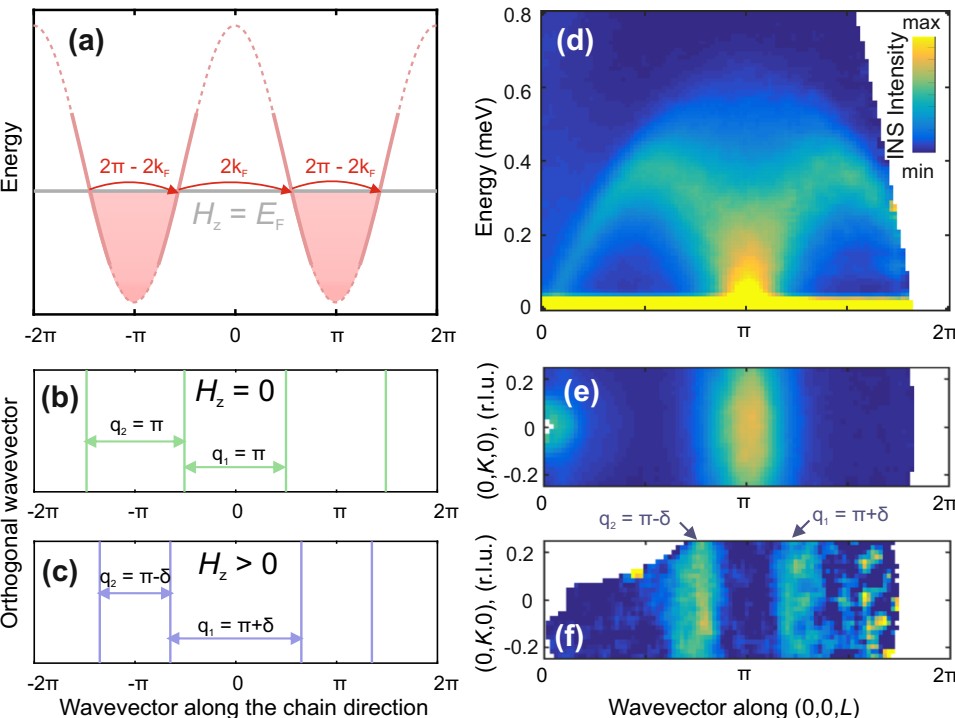

**Fig. 1 Bandstructure of fermions in the Heisenberg chain and its manifestations in the INS spectrum of YbAlO₃. a** Red line shows the bandstructure in the fermionic representation for a Heisenberg spin chain. Part of the curve is dotted to indicate that the linearization is strictly valid only in the vicinity of the chemical potential. The horizontal gray line shows the positions of the field-dependent chemical potential, $E_F = H_z$; the red shaded area shows the occupied part of the fermionic band. 1D Fermi surfaces for $H_z = 0$ and $H_z > 0$ are shown by vertical lines in **b** and **c**, respectively. Arrows in **b, c** indicate the wavevectors of the $2\mathbf{k}_F$ Kohn instability. **d** Inelastic neutron scattering (INS) spectrum of YbAlO₃ collected at $T = 1$ K, $B = 0$ T (1D regime of decoupled chains), showing the spinon continuum. **e, f** Constant-energy slices in the ($0KL$) scattering plane measured at $T = 1$ K, $B = 0$ T (**e**); $T = 0.05$ K, $B = 0.6$ T (**f**). The signal is integrated just above the elastic line, over the interval $\hbar\omega = [0.05−0.1]$ meV. The INS signal shows stripes at the wavevector of the Kohn anomaly, $q = \pi$ at $B = 0$ and $q = \pi \pm \delta$ at $B = 0.6$ T.

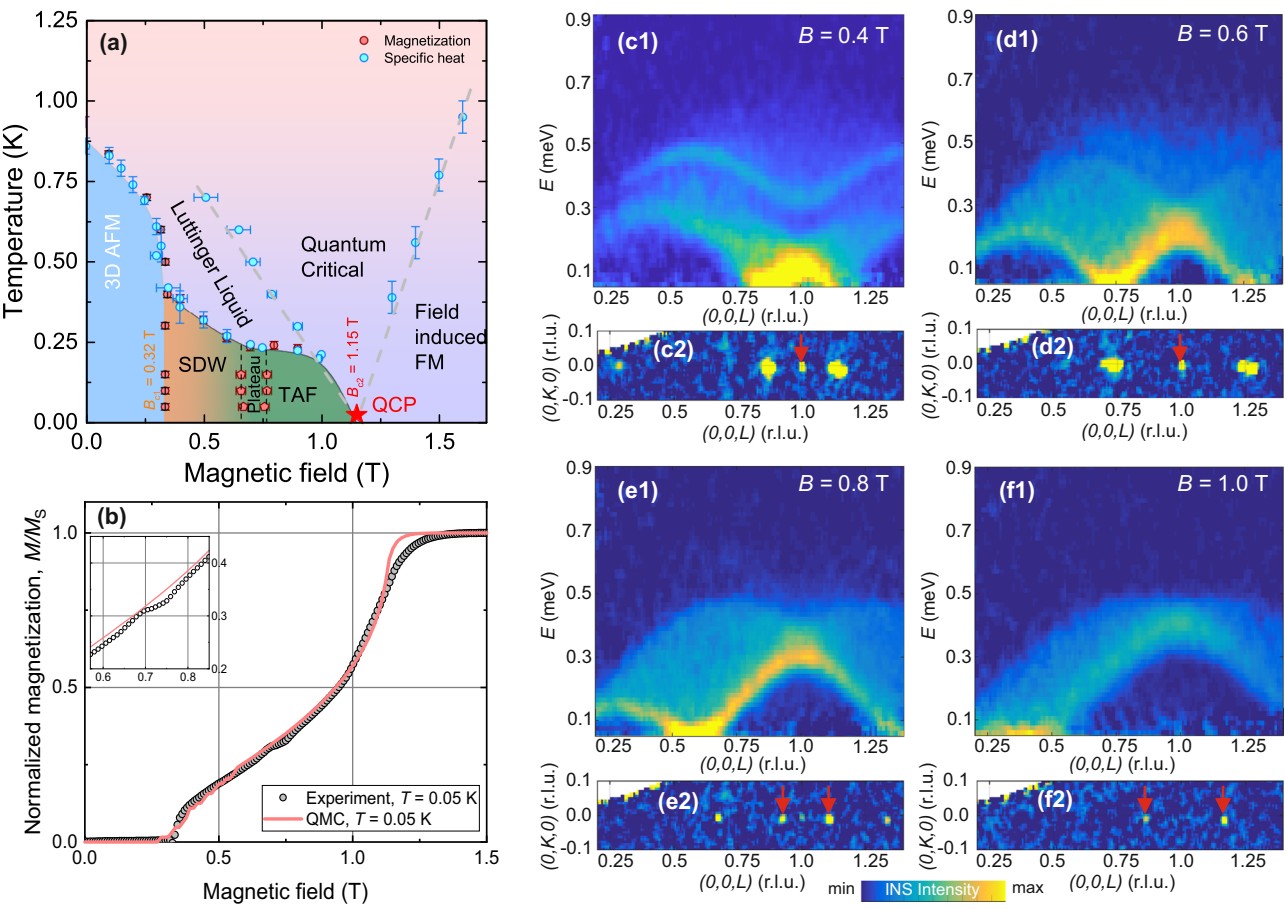

**Fig. 2 Phase diagram and magnetic excitations in YbAlO₃. a** Phase diagram of YbAlO₃ reconstructed from specific-heat and magnetization measurements for $B \parallel [100]$. Colored areas show three magnetically ordered phases; red points at low temperatures and $B \approx 0.7$ T mark the position of the $M_S/3$ plateau. Gray dotted lines indicate crossovers between Luttinger liquid, quantum critical (QCP), and field-polarized regimes. Horizontal and vertical error bars correspond to uncertainties in the determination of the peak positions for constant-temperature and constant-field scans, respectively. **b** Normalized magnetization, $M/M_S$, measured at $T = 0.05$ K, and magnetization curves calculated using (QMC). The red curve shows noisy features within the spin-density wave (SDW) phase due to finite-size errors in the calculations. (**c1–f1**) Energy-momentum cuts along the (00L) direction measured at $T = 0.05$ K and in different magnetic fields (shown in each panel). All spectra show no gap within the experimental resolution of 50 μeV. Because the dispersion of magnetic excitations in YbAlO₃ has 1D character, the data were integrated over the whole available range in the orthogonal directions ($K = [-2; 2]$ r.l.u. and $H = -[0.3; 0.3]$ r.l.u.). (**c2–f2**) ENS maps within the (0KL) plane obtained by integrating the same dataset as in **c1–f1** around the elastic position, $E = [-0.05; 0.05]$ meV with $H = [-0.05; 0.05]$. Red arrows indicate extrinsic Bragg reflections due to the presence of the twin in the sample, which is discussed in detail in Sec. (S1) of the SI. Subtraction of the 2 T data set was applied to all spectra in **c1–f2**.

in Fig. 1d. The finite dipolar interchain coupling with an Ising-like anisotropy[19] causes a commensurate AFM order below $T_N = 0.88$ K. The field-temperature phase diagram is summarized in Fig. 2a. Application of a longitudinal magnetic field ($B \parallel a$) suppresses the commensurate ordering at low temperatures in favor of an incommensurate (IC) phase [Fig. 1e, f] at $B_{c1} = 0.32$ T, and drives a quantum phase transition to the field-polarized phase at $B_{c2} = 1.15$ T. Moreover, the magnetization curve shows a weak plateau close to $m = 1/3$, as shown in Fig. 2b. We note that the observed incommensurate SDW order in YbAlO₃ is surprising, because the low-energy physics of a Heisenberg chain is described by a Tomonaga-Luttinger liquid (TLL) in which the $2k_F$ scattering is irrelevant at finite fields. Although it has been recognized that the Ising anisotropy of the interchain coupling has a crucial role[20,21], this field-induced SDW order in YbAlO₃ remains poorly studied.

In this work, we have performed elastic and INS measurements to reveal the details of magnetic ordering within the low-temperature SDW phase of YbAlO₃. We found that the combination of isotropic intra- and Ising-like interchain interactions gives rise to the consecutive formation of the SDW and transverse

antiferromagnetic (TAF) phases as the longitudinal magnetic field is increased. The interchain coupling within the SDW produces an effective incommensurate potential, which opens a way to multiple scattering of the fermions (the analogy of Umklapp process in metals and graphene[22–24]). This results in the formation of satellites at multiples of the incommensurate fundamental wavevector, $n\mathbf{q}$, which we have observed in the neutron diffraction experiments. To verify the physical picture of multiple scattering we have performed detailed numerical calculations by density matrix renormalization group (DMRG) on quasi-1D systems and by QMC on the 3D coupled-chain model.

## Results and discussion
We begin the presentation of our results with the INS spectra measured in the IC phase using Cold Neutron Chopper Spectrometer (CNCS). Figure 2c1–f1 shows energy-momentum slices along the (00L) direction taken at base temperature, $T = 0.05$ K, and $B = 0.4$, 0.6, 0.8, and 1 T. The spectra consist of broad, gapless continua with a bandwidth of $\approx 0.5$ meV, which evolve

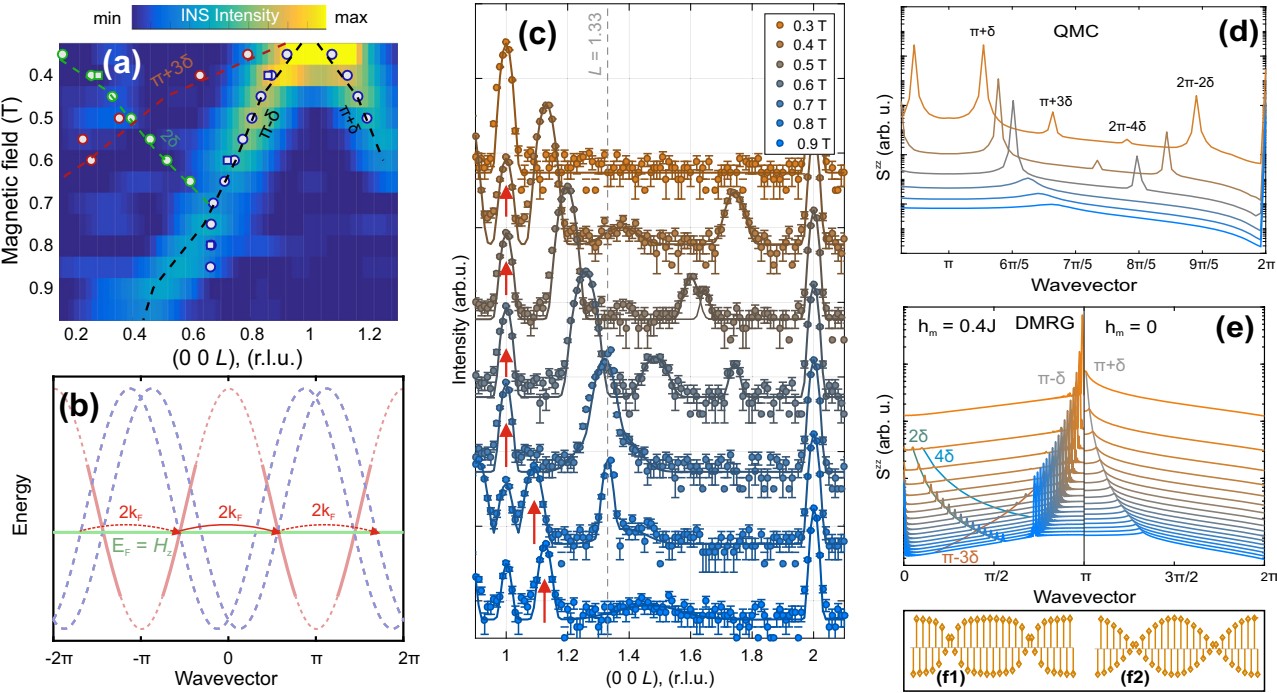

**Fig. 3 Magnetic diffraction and origin of the satellites. a** Field-dependence of $S(\mathbf{q}, \omega = 0 + \varepsilon)$ measured on CNCS at $T = 0.05$ K. The color map shows the quasielastic signal along the $(00L)$ direction. The data are integrated within $H = [-0.1; 0.1]$ (r.l.u.); $K = [-0.15; 0.15]$ (r.l.u.) and $E = [30; 70]\mu$eV. Blue, green and red dots show positions of Bragg peaks extracted from the FLEXX (circles) and CNCS (squares) measurements; corresponding dotted lines show $\pi + \delta(B)$, $2\delta(B)$, and $\pi + 3\delta(B)$ splittings extracted from the soft-mode fitting. **b** The red line shows the bare bandstructure in the fermionic representation for a Heisenberg spin chain in a finite magnetic field, whereas the blue dotted lines display two folded bands caused by $h_{ic}$. The green horizontal line shows the position of the chemical potential. Red arrows indicate paths for multiple scattering. Note that the small gap at the Fermi level is not shown. **c** Representative ENS scans along the $(00L)$ direction measured on FLEXX at $T = 0.05$ K at different magnetic fields. Red arrows mark extrinsic reflections due to the twin. **d** $S^{zz}(\mathbf{q}, \omega = 0)$ within the spin-density wave (SDW) phase calculated using quantum Monte–Carlo (QMC). **e** $S^{zz}(\mathbf{q}, \omega = 0)$ calculated by density-matrix renormalization group (DMRG) method for different external magnetic fields with (left) and without (right) the incommensurate field, $h_{ic}$. One can see that $h_{ic}$ stabilizes the ordering at the fundamental wavevector $2k_F$ and produces additional harmonics at $2nk_F$. **f1–f2** Distorted SDW with square-like modulation calculated by DMRG (**f1**) compared with a harmonic SDW (f2). The distorted SDW in **f1** was calculated using Eq. (3) with $h_{ic} = 0.4J$. Note that $h_{ic}$ exceeds the real internal field and was chosen to demonstrate more clearly the nature of the distortions caused by the multiple scattering processes at $\mathbf{Q} = n\mathbf{q}$. The intensity axes in **c**–**e** are shown on a logarithmic scale and the curves are shifted vertically for ease of viewing. The vertical error bars in **c** represent statistical errors of one s.d.

with a magnetic field. Previously, it was shown that the essential features of the spectra obtained at $B \geq 0.6$ T can be reproduced by a $S = 1/2$ Heisenberg spin-chain model in a magnetic field without introducing any kind of interchain coupling, $J_{ab}$[17]. This is because the interchain coupling has a dipolar origin, and is therefore proportional to the ordered moment, which is suppressed with increasing field. Only the spectrum collected at $B = 0.4$ T shows qualitatively new features at high energies and the introduction of interchain coupling is essential to obtain its correct description, but $J_{ab}$ affects the low-energy part of the spectrum, which was used to extract $2k_F$ only very weakly. Thus, by examining the spectra at $\hbar\omega \approx 0 + \varepsilon$ ($\varepsilon$ is a small number, on the order of the spectrometer resolution) we can gain information about the quasi-zero-energy response of 1D subsystem as shown in Fig. 1e, f. However, even though $J_{ab}$ is weak, it is an essential ingredient of the system, because it induces the 3D long-range order, which manifests itself by the appearance of sharp Bragg peaks in the elastic channel.

Let us turn to the description of the field-induced magnetic order. As discussed in the introduction, the magnetic field effectively shifts the chemical potential of the fermion band and induces a nesting of the FS at $q_L = 2k_F = \pi(1 \pm 2m)$. In the spectrum, this instability corresponds to the zero-energy position of the longitudinal mode, $S^{zz}(2k_F, \omega = 0 + \varepsilon)$, which we call the soft mode throughout the text. Within the SDW phase at $B = 0.4$ and

0.6 T, the soft-mode positions, $2k_F = 1 \pm 0.13$ and $1 \pm 0.28$ (r.l.u.), respectively, can be extracted from the inelastic spectrum (see details of the soft-mode fitting in Sec. (S3) of the Supplementary Information (SI)) and the elastic Bragg peak is located exactly at the same position, $q_{L1} = 1 \pm \delta_1$. When the field is increased to 0.8 T, the zero-energy mode position, $2k_F = 0.59$ (r.l.u.), shifts continuously away from (001), but the position of the elastic satellite, $q_{L1} = 0.66$ (r.l.u.), deviates from $2k_F$. At $B = 1$ T, the first set of satellites disappears, whereas the zero-energy mode position shifts gradually towards the ferromagnetic (FM) (002) and (000) positions. We summarize the evolution of the soft mode as a function of the magnetic field in Fig. 3a. The background-subtracted INS cuts along the $(00L)$ direction, integrated just above the elastic line, are plotted at different magnetic fields so that one can see the almost linear evolution of the soft-mode position, $2k_F$.

Note that some part of the intensity remains almost unchanged at the (001) position up to 0.75 T and then exhibits a second splitting, $\delta_2$. The field dependence of $\delta_2$ is similar to $\delta_1$. However, the second splitting, as well as the overall behavior of the (001) peak, is not intrinsic to the spin-chain physics of YbAlO$_3$, but rather is caused by the small degree of twinning in the sample, with the second twin rotated by 90° with respect to the main crystalline axes and oriented with [010]∥$B$ (see Sec. (S1) of the SI for a detailed analysis of the second splitting). Henceforth, we will

not consider the second splitting and comment only briefly on the field-induced evolution of the (001) peak.

To further reveal the details of the magnetic structure of the IC phase, we performed an elastic neutron scattering (ENS) experiment using the triple-axis spectrometer (TAS) FLEXX[25] and the primary results of our measurements are summarized in Fig. 3a, c. Figure 3c shows the raw ENS scans measured along the (00L) direction as a function of magnetic field at $T = 0.05$ K. One can see that below the first critical field, $B_{c1}$, the diffraction patterns contain commensurate AFM and structural reflections at (001) and (002), respectively. Above $B_{c1}$, the AFM peak splits into $q_{L1} = 1 \pm \delta_1$, and $\delta_1$ follows the position of the soft mode of the spectra and magnetization curve up to the 1/3 plateau. Surprisingly, we found that, in addition to the primary IC satellites at $q_{L1}$, the diffraction pattern exhibits two additional weak peaks, whose positions evolve with magnetic field as $q_{L2} = 2 \pm 2\delta_1(B)$ and $q_{L3} = 1 \pm 3\delta_1(B)$, as shown in Fig. 3c. The intensities of those peaks are between 100 and 1000 times weaker than the structural and primary IC peak. For this reason we were able to resolve only the strongest harmonics of $q_{L2}$ at $B = 0.4$ T in our CNCS measurements, visible as the weak peak at (0 0 0.27) r.l.u. in Fig. 2c2.

At $B = 0.7$ T, the $q_{L1}$ and $q_{L2}$ peaks merge at (0 0 $\frac{4}{3}$) and the ENS signal can be deconvolved using two peak functions: (i) a sharp peak, whose position is locked at $q_L = 1.33$ (r.l.u.) and whose intensity is suppressed gradually with field, until it finally vanishes above 0.85 T; (ii) a broad peak, which corresponds to the soft-mode contribution and shifts continuously with magnetic field. We summarize the field dependence of the IC peaks along with the soft-mode position in Fig. 3a. The locking of the Bragg peak position takes a place around the same field range where the $M_S/3$ plateau was observed. In spite of the presence of the soft mode at finite wavevector, $\mathbf{q} = 2\mathbf{k}_F$, in the field range between 0.9 T and $B_{c2}$, the sharp incommensurate peak is absent, indicating that the SDW phase is suppressed, although the sample remains in an ordered phase, as is evident from the sharp features in thermodynamic measurements[17]. This suggests that the magnetic ordering changes to the transverse channel, which supports a gapless Goldstone mode as a consequence of the broken in-plane $U(1)$ symmetry. The soft mode in the longitudinal channel then becomes overdamped and fades out with increasing field.

Indeed, a TAF phase with nonzero transverse component of the spin structure factor, $S^{xx}(\mathbf{q} = \pi)$, for $H_z \gtrsim 0.7$ T is confirmed in both (QMC) and DMRG calculations[20,21]. The TAF phase is continuously vanishing at $B_{c2} = 1.15$ T. Unfortunately, owing to the presence of the small twin in our sample, we could not compare the experimental field-dependence of the peak at $\mathbf{q} = \pi$ quantitatively with the theoretical predictions, but qualitatively both follow the same trend and gradually vanish close to $B_{c2}$ (see Sec. (S1) of the SI for an analysis of the (0 0 1) peak intensity).

The origin of the satellites can be understood as a competition of intrachain Heisenberg and interchain Ising-like interactions. The latter enhance longitudinal spin correlations and therefore support a collinear ordering with maximal $\langle S^z \rangle$. On the other hand, at the finite field, the Heisenberg term promotes ordering in the transverse components. The compromise between these energy scales can be achieved by the formation of an incommensurate SDW with a weak modulation at integer multiples of the fundamental propagation vector $n\mathbf{q}$.

To better understand this phenomenon, we first map the Heisenberg spin chain in a magnetic field (Eq. (1) with $\Delta = 1$) to a system of interacting fermions (full details can be found in Sec. (S4) of the SI). The low-energy physics is dominated by the left- and right-moving fermions, respectively, $\Psi_L$ at $-k_F$ and $\Psi_R$ at $k_F$. Then, following the standard bosonization procedure (see Sec. (S4) of the SI and ref. [17]), the physics of these fermions can be understood by the following Luttinger model

$$\mathcal{L} = \frac{1}{2\pi K}\left[\frac{1}{u}(\partial_\tau \phi)^2 + u(\partial_x \phi)^2\right], \quad (2)$$

where $u$ and $K$ are the spin-wave velocity and the Luttinger parameter, respectively. This model describes a TLL with dominant transverse correlations. In terms of fermions, it shows that in addition to the band renormalization effect, the intrachain Ising interaction is irrelevant at low energies. However, the Ising term of the interchain coupling, which is introduced as an effective contribution to the field term, produces scattering at $q = 2k_F$ between the left- and right-moving fermions [Fig. 3b]. This Umklapp process adds a term $J_{ab} \cos(2\phi)$ to Eq. (2), meaning that it is a relevant perturbation of the Luttinger model, and its consequence is the band-folding of the left- and right-moving fermions with the incommensurate wavevector $2k_F$ shown in Fig. 3b. In 3D coupled chains, this instability of FS gives rise to an SDW order at the incommensurate wavevector $\mathbf{q} = 2\mathbf{k}_F$ and opens a small gap at the Fermi level. Moreover, the $n$th order perturbation in $J_{ab}$ create processes involving multiple coherent (Umklapp) scattering of fermions with incommensurate wavevectors $n\mathbf{q} = 2n\mathbf{k}_F$, which accounts for the observed satellite peaks in the ENS measurements.

To further demonstrate that the bosonization analysis describes the physics for the satellites, we perform numerical simulations on microscopic spin Hamiltonians. We first use DMRG to study an effective single-chain spin model,

$$\mathcal{H} = \sum_i (JS_i S_{i+1} - S_i^z(H_z + h_{ic}\cos(2k_F r_i))). \quad (3)$$

In this equation, the second term shows the effect of the external magnetic field while $h_{ic}$ denotes the interchain incommensurate "molecular" field produced by the neighboring chains; $2k_F$ is determined for each $H_z$ as $2k_F = \pi(1 - 2m)$. $S^{zz}(\mathbf{q}, \omega = 0)$ calculated for different $H_z$ in the presence and absence of modulated fields is shown in Fig. 3e. It is clear that, in absence of the incommensurate modulation, $S^{zz}$ shows only a broad feature at the fundamental wavevector, $\mathbf{q} = 2\mathbf{k}_F$, whereas $h_{ic} = 0.4 J$ produces a series of additional satellites at $n\mathbf{q}$. The resulting distortion of the SDW is represented in Fig. 3f1 has a pronounced "square"-like shape when compared with the harmonic SDW [Fig. 3f2]. We note that the strength of internal field $h_{ic}$ used in DMRG calculations was overemphasized, compared with its quantitative value in YbAlO$_3$, in order to present the type of distortion more clearly.

Finally, to show that the molecular-field approximation used in our DMRG modeling is justified, we have performed QMC simulations on a 3D array of Heisenberg spin chains coupled weakly by almost Ising-like FM interactions (see Sec. (S5) of the SI for details of the model). It has been shown that this model reproduces the phase diagram of YbAlO$_3$ and exhibits three distinct ordered phases: low-field commensurate AFM, longitudinal SDW, and transverse canted AFM[21,26], although it does not reproduce the $M_S/3$ plateau (the plateau was not the primary focus of the present work, and we use DMRG to discuss its possible origin in Sec. (S6) of the SI). Figure 3d shows the calculated longitudinal spin structure factors, $S^{zz}(\mathbf{q}, \omega = 0)$, within the SDW phase at several representative magnetic fields. One can see that the signal consists of four peaks at $\mathbf{q}_1 = \pi\delta$, $\mathbf{q}_2 = 2\pi - 2\delta$, $\mathbf{q}_3 = \pi + 3\delta$, and $\mathbf{q}_4 = 2\pi - 4\delta$, or in general $n\mathbf{q}_1$. Note that the intensity of the fundamental peak at $\mathbf{q}_1$ significantly exceeds the weaker satellites at $\mathbf{q}_2$ and $\mathbf{q}_3$, in good agreement with both DMRG results and the experimental observations.

Therefore, by combining the results of bosonization, DMRG and QMC we can conclude that the Umklapp scattering acts as an additional incommensurately modulated Zeeman potential of the

form $J_{ab}\cos(2k_F r)$ in a single chain. This potential modifies the fermion bandstructure, as shown in Fig. 1a, and allows the multiple coherent scattering of fermions represented schematically in Fig. 3b, which gives rise to the observed satellites. In real space, these processes distort the SDW towards the more square-like shape shown in Fig. 3f1. It is worth noting that this mechanism for stabilizing higher-order harmonics should be more prominent with larger intrachain Ising anisotropy ($\Delta > 1$), and thus one would expect that the satellites also appear in $BaCo_2V_2O_8$ and $SrCo_2V_2O_8$[15,16,27]. However, because $S^{zz}(n\mathbf{q_1}) \propto (J_{ab}/J)^{\bar{n}}$, it may be difficult to resolve the satellites in ENS data because of the weak $J_{ab}/J$ ratio ($\lesssim 1/30$) in these compounds.

The physical consequences of multiple fermion scattering should manifest not only in reciprocal space-sensitive experiments, such as neutron diffraction, but also in low-temperature transport and thermodynamic properties. Umklapp scattering is known to suppress the thermal and electrical conductivity in metals, and thus multiple fermion scattering should provide an additional scattering channel reducing the thermal conductivity mediated by spin excitations. The presence of the incommensurate internal field in $YbAlO_3$ that allows for multiple scattering also opens a small gap in the spin excitation spectra, thus the spinon thermal conductivity of $YbAlO_3$ within the SDW phase would be altered in two ways. This interplay between the gapping effect and the extra contribution to the fermionic component of multiple scattering makes thermal transport measurements on $YbAlO_3$ highly desirable. The presence of the gap should also be observable by specific-heat measurements, which are similarly sensitive to the density of states of the magnetic quasiparticles. However, we note that a direct experimental detection in either quantity may be difficult, because the strength of multiple scattering scales with $(J_{ab}/J)$, and therefore is rather weak.

To summarize, we have observed distorted SDW ordering in $YbAlO_3$ by means of neutron scattering and have provided a detailed theoretical description. These results are of interest because we show how diffraction can be used to probe the bandstructure of fermionic quasiparticles in quantum magnets. We have demonstrated that the incommensurate effective field modulates the chemical potential and produces additional shadow bands as shown in Fig. 3b, which allow multiple (Umklapp) scattering of the fermions. To our knowledge, the coherent multiple scattering of fermions has not been detected before in either quantum magnets or one-dimensional metals. Thus, our results provide valuable insight into coherent quantum many-body phenomena and motivate further efforts to search for this effect in low-dimensional charge- and SDW condensed-matter systems, as well as in ultracold atoms in optical lattices.

## Methods

**Experimental details**. A single crystal of $YbAlO_3$ with a mass of $\approx 0.5$ g was prepared by a Czochralski technique, as described elsewhere[28,29]. The INS measurements were performed at the time-of-flight spectrometer CNCS[30,31] at the Spallation Neutron Source (SNS) at Oak Ridge National Laboratory with $E_i = 1.55$ and 3.32 meV. To generate the magnetic field we used an 8 T vertical magnet. ENS measurements were performed using the TAS FLEXX at the Helmholtz Zentrum Berlin[25] with fixed $\mathbf{k}_i = \mathbf{k}_f = 1.3$ Å$^{-1}$. To increase the q-resolution we installed a 60' Söller collimator between the sample and the analyzer, and set both monochromator and analyzer with no horizontal focusing. The magnetic field was applied using the vertical cryomagnet VM-4. In both the neutron scattering experiments, the sample was oriented in the $(0KL)$ scattering plane and the magnetic field was applied along with the easy $a$ axis. The magnetization was measured using a high-resolution Faraday magnetometer[32] with a dilution cryostat. Reduction of the TOF data was performed using the Mantid[33] and Horace[34] software packages.

**QMC and DMRG calculations**. QMC simulations of the coupled spin-chain model were performed with a maximum system size of $20 \times 20 \times 128$ sites and the lowest temperature accessed was $T/J = 0.01$. DMRG calculations were performed for two different spin models. The first one is given by Eq. (3) for the calculation of $S^{zz}(\mathbf{q}$,

$\omega = 0)$ we used the standard DMRG method with open boundary conditions. We studied a 1D chain of $L = 240$ and 1600 density-matrix eigenstates that were kept in the renormalization procedure. In this way, the maximum truncation error, i.e., the discarded weight, can be negligible ($\leq 10^{-16}$). The second model was a two-leg spin-ladder with a next-neighbor frustrating intrachain interaction, which was used to provide a qualitative description of the $M_S/3$ plateau, see Sec. (S6) of SI.

## Data availability
The data that support the findings of this study are available from the corresponding authors upon reasonable request.

## Code availability
The code that supports the findings of this study is available from the corresponding authors upon reasonable request.

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

## Acknowledgements
We acknowledge C.D. Batista for fruitful discussions, U. Nitzsche for technical assistance, P.Y. Portnichenko for help with the neutron-scattering experiment, and B. Normand for insightful suggestions and careful proofreading of the manuscript. This research used resources at the Spallation Neutron Source, a DOE Office of Science User Facility operated by Oak Ridge National Laboratory. S.N. acknowledges support by the Deutsche Forschungsgemeinschaft through SFB 1143 project no. A05. S.E.N. and A.S.S. acknowledge support from the International Max Planck Research School for Chemistry and Physics of Quantum Materials (IMPRS-CPQM). The work at Renmin University was supported by the Ministry of Science and Technology of China, National Program on Key Research Project (grant number 2016YFA0300504), the National Science Foundation of China (grant number 11674392), and the Fundamental Research Funds for the Central Universities and the Research Funds of Renmin University of China (grant number 18XNLG24). J.W. acknowledges additional support from a Shanghai talent program. The work at Shanghai Jiao Tong University is sponsored by the Natural Science Foundation of Shanghai with grant no. 20ZR1428400 and Shanghai Pujiang Program with grant no. 20PJ1408100 (J.W.).

## Author contributions
S.E.N., A.S.S., J.X., L.S.W., N.S.P., and A.P. performed neutron-scattering experiments. S.E.N. and M.B. carried out the magnetization measurements. L.V. provided the single crystals used in this study. S.N. performed the DMRG calculations and Y.F., and R.Y. performed the QMC calculations. S.N., R.Y., and J.W. developed the theoretical framework. R.Y., J.W., and A.P. supervised and supported the study. All authors discussed the results and contributed to the writing of the manuscript.

## Funding

## Competing interests
The authors declare no competing interests.
