## [Peer Review File · Nature Communications]

REVIEWER COMMENTS

Reviewer #1 (Remarks to the Author):

This manuscript analyzes details of the field-induced magnetic order and excitations in the compound YbAlO_3 , which is a remarkable realization of almost perfect Heisenberg spin-1/2 antiferromagnetic chains with magnetic moments originating from rare-earth atoms. The current team has investigated this compound extensively in several previous work, including a detailed neutron scattering investigation in 2019. The current work advances on one particular front, which is a deeper understanding of the field induced order using careful neutron diffraction, elastic and inelastic neutron scattering, torque magnetization and an equally impressive arsenal of theoretical techniques ranging from field theory to DMRG and QMC. Compared to previous work, and as far as I can see, the main discovery is that a non-harmonic spin-density wave order, followed by a 1/3 magnetization plateau phase, is stabilized in this compound. The stabilization of that order is interesting and emerges from peculiar Ising-like inter-chain coupling (ultimately of dipolar origin). Accompanying the main spin-density wave peak, which incommensurate position shifts as expected when the system is magnetized, the authors discovered two higher-harmonic reflections which essentially reflect the « squaring-up » of the longitudinally polarized spin density wave. In turn, this originates from the inter chain-coupling. The crux of the work is to deploy the usual mapping of the spin-chain into a system of interacting fermions (Jordan Wigner) to understand the effect of interchain coupling as Umklapp scattering processes for these fermions. I think this conclusion, while relatively narrow in its current implications, is interesting.

When it comes to my recommendation, I would like to answer three questions: (1) is this work technically sound?, (2) is it interesting enough for Nature Communications?, (3) can the manuscript be improved?

For (1), I think the work is of high quality, sound and expertly conducted. Of course, there is a complication with a small twin in the sample that produces spurious peaks. My experience is that these things happen and while they should be avoided sometimes there is no way around it. For instance that sample twin may not be detectable by X-ray (Laue or Single-Crystal) and even if seen in neutron diffraction, it might be unclear which part of the crystal is responsible. Given that the twin is well understood, my judgement is that this work is technically of high quality.

For (2), my answer is Perhaps. I think the scope of the work is relatively limited. On the one hand, one thing that I like is that the paper holds water without resorting to too many references to the same team's previous works. But looking at previous work from this team on YbAlO_3 , it looks like almost all of it was previously known (with the exception of the squaring of the SDW).

For (3), I think mapping this problem back to fermions is interesting and this is where the impact of this work really is. There are two aspects that could be improved in that regard and that may alleviate my concern (2).

— First, It is not immediately clear to me how the particular form of the inter-chain coupling leads to the reconstruction shown in Fig. 3(b). I understand the equation but is there a simple cartoon that could explain it?

— Second, and perhaps most importantly, can the effect of Umklapp scattering between spinons be seen in experiments that probes their transport or other thermal properties? One of the amazing thing about a critical system like here is that the heat-capacity has a residual gamma-term, so I am wondering if the fingerprint of Umklapp processes can be see from thermal transport measurements on this sample? Even if experiments cannot be done, perhaps predictions on how transport and thermal properties of YbAlO₃ are expected to be impacted by this discovery of Umklapp processes would be quite informative and kind of « close the loop »

Reviewer #2 (Remarks to the Author):

In this paper, the authors have found satellite Bragg peaks beside the main peak of the incommensurate order for the Heisenberg chains coupled with anisotropic inter-chain couplings, using the neutron diffraction experiment. Also, reliable theoretical analyzes including numerical simulations clarified that the satellite peaks are cooperatively induced by the magnetic field and the anisotropic inter-chain couplings.

I think that this finding has clarified a novel aspect of the field induced orders for quantum spin systems. In the present version of the manuscript, however, I think that some explanations based on the fermion picture, which are especially emphasized in this paper, sound slightly misleading. In particular, the band structure in Fig. 1(a) and Fig. 3(b) is that of the free fermion(XY chain), not that of the Heisenberg chain.

Actually, the spin wave velocity of the Heisenberg chain is nonperturbatively

renormalized to $\pi J/2$ from J due to the interaction ($S^z S^z$ -term), which leads the effective coupling of $u = \pi J$ in the bosonic field theory (2) in which the intra-chain couplings were already involved. Also, the inter-chain coupling of Eq. (5) is based on the effective bosonic field of Eq. (2). I therefore think that although the theoretical and numerical calculations are correct, the free-fermion-based explanation in this paper should be properly modified.

Also, the inter-chain interaction effect on the dynamical spin structure factors of 1D quantum spin chains are well described by the Zeeman ladder approach:

Prog. Theor. Phys. 63, 743 (1980)

Phys. Rev. Lett. 114, 017201 (2015)

I think that these papers should be included in the references.

Then, the authors should describe how the satellite Bragg peaks appear in AbAlO_3 beyond the Zeeman ladder approach.

In a QMC simulation for a finite chain length L , the wavenumber in the chain direction is generally quantized to be $2\pi/L \times (\text{integer})$, which may not be compatible with the incommensurate wavenumber associated with the magnetic field.

I hope that the authors provide some comment about this finite size effect for $S(q, \omega)$.

Response to Referee Comments and Consequent Revisions of Manuscript

Referee questions, remarks or comments are in black, our replies are in red.

1st Referee: This manuscript analyzes details of the field-induced magnetic order and excitations in the compound YbAlO_3 , which is a remarkable realization of almost perfect Heisenberg spin-1/2 antiferromagnetic chains with magnetic moments originating from rare-earth atoms. The current team has investigated this compound extensively in several previous work, including a detailed neutron scattering investigation in 2019. The current work advances on one particular front, which is a deeper understanding of the field induced order using careful neutron diffraction, elastic and inelastic neutron scattering, torque magnetization and an equally impressive arsenal of theoretical techniques ranging from field theory to DMRG and QMC. Compared to previous work, and as far as I can see, the main discovery is that a non-harmonic spin-density wave order, followed by a 1/3 magnetization plateau phase, is stabilized in this compound. The stabilization of that order is interesting and emerges from peculiar Ising-like inter-chain coupling (ultimately of dipolar origin). Accompanying the main spin-density wave peak, which incommensurate position shifts as expected when the system is magnetized, the authors discovered two higher-harmonic reflection which essentially reflect the «squaring-up» of the longitudinally polarized spin density wave. In turn, this originates from the inter chain-coupling. The crux of

the work is to deploy the usual mapping of the spin-chain into a system of interacting fermions (Jordan Wigner) to understand the effect of interchain coupling as Umklapp scattering processes for these fermions. I think this conclusion, while relatively narrow in its current implications, is interesting.

We thank the Referee for his or her careful review of the manuscript, for several insightful comments and for recognizing the interest of our results.

1st Referee: *When it comes to my recommendation, I would like to answer three questions: (1) is this work technically sound?, (2) is it interesting enough for Nature Communications?, (3) can the manuscript be improved?*

For (1), I think the work is of high quality, sound and expertly conducted. Off course, there is a complication with a small twin in the sample that produces spurious peaks. My experience is that these things happen and while they should be avoided sometimes there is no way around it. For instance that sample twin may not be detectable by X-ray (Laue or Single-Crystal) and even if seen in neutron diffraction, it might be unclear which part of the crystal is responsible. Given that the twin is well understood, my judgement is that this working is technically of high quality.

We are grateful for the Referee's positive evaluation of our analysis to isolate the contribution of the twin to the diffraction signal. Indeed, we carefully scanned the sample using X-ray Laue diffraction, but were not able to identify the position of the twin. We further note that according to the available literature about the crystal growth of orthorhombic aluminates with perovskite structure, the twinning problem is rather common. The degree of twinning depends not only on the crystal growth conditions, but also on the perovskite structural deformation - the smaller is the deviation from a cubic perovskite aristotype, the higher is twinning of the crystal. To the best of our knowledge, there is no information about the twinning problem in YbAlO_3 , although we expect it to be similar to the case of YAlO_3 , in which separate microscopic twin lamella were often observed (D.I. Savytskii, J. Cryst. Growth 209, 874-882 (2000)). Assuming that the phase separation takes place on the micron lengthscale, it would hardly be possible to obtain a macroscopic twin-free sample for neutron scattering from the currently available crystals, and an improvement in growth technology would be necessary to avoid this phase separation.

1st Referee: *For (2), my answer is Perhaps. I think the scope of the work is relatively limited. One the one hand, one thing that I like is that the paper holds water without resorting to too many references to the same team's previous works. But looking at previous work from this team on YbAlO_3 , it looks like almost all of it was previously known (with the excepting of the squaring of the SDW). For (3), I think mapping this problem back to fermions is interesting and this is where the impact of this work really is.*

We agree with the Referee that many general aspects of the magnetic behaviour of YbAlO_3 have already been addressed in previous publications, and we are glad that the Referee appreciated our best efforts to avoid any unnecessary repetitions that were not directly required to focus on the primary result of the present paper. In this work, we report two experimental observations, which concern the field-induced evolution of the magnetic structure:

- (i) We found that the SDW phase is stabilized only within a limited field range, rather than throughout the whole region between B_{c1} and the QCP. According to recent QMC calculations, the transition between the SDW and TAF phases is controlled by the spin anisotropy of the inter-chain interaction [Y. Fan and R. Yu, Chin. Phys. B Vol. 29, No. 5 (2020) 057505].
- (ii) We resolved the second and the third harmonics of the primary incommensurate satellites within the SDW phase.

We also agree that the fermionic interpretation of the observed satellites is one of the most exciting outcomes that one may deduce from our results. It provides an intuitive understanding for the formation of the incommensurate SDW phase and reveals a clear physical mechanism for multiple fermion scattering, which is responsible for the formation of satellites, as we tried to emphasize in the original submission. Having said this, we would disagree with the Referee that “almost all of it was previously known”, because, to the best of our knowledge, the primary outcome of our work, the multiple coherent scattering of fermions, was never reported before. Now, following the suggestion of the Referee, we have included a further discussion of some more general physical manifestations of multiple scattering, in transport and thermodynamic properties, as detailed below.

1st Referee: *There are two aspects that could be improved in that regard and that may alleviate my concern (2). First, It is not immediately clear to me how the particular form of the inter-chain coupling leads to the reconstruction shown in Fig. 3(b). I understand the equation but is there a simple cartoon that could explain it?*

We thank the referee for this question, which helps to improve the conceptual presentation of the key physics. As we already addressed in the manuscript and the supplementary information, the interchain coupling causes the Umklapp process that transfers a momentum of $q = 2k_F$. This is seen most transparently in the fermionic representation, where the interchain Hamiltonian is approximated as $H_{ic} \approx J_{ab}(q = 2k_F) \Psi_L^\dagger \Psi_R$ at low energies, where $\Psi_{L/R}$ is the single-particle operator of left/right-moving fermions and $J_{ab}(q)$ the strength of the Umklapp scattering. This scattering leads to the folding of the bands with nesting wavevector $q = 2k_F$ (or, equivalently, the folding of the Brillouin zone). Two of the folded bands (usually denoted as shadow bands) are shown as the dashed curves in Fig. 3(b). This band-folding effect occurs quite generally in a large number of systems with SDW or CDW order. However, the situation we address

in YbAlO_3 is unique for two reasons: first, the SDW order is induced by the interchain coupling and not the intrachain interactions; second, the nesting vector q is incommensurate. As a consequence of the second point, nq is also incommensurate for any integer n , and thus there is an infinite series of folded bands in the Brillouin zone. This makes multiple fermion scattering active at all wavevectors nq , and it is responsible for the observed satellite peaks. We wish we could provide a simpler cartoon, but given the complication introduced by the incommensurate wavevector, we still find that Fig. 3(b), as a graphical representation of the meaning of H_{ic} above, gives the clearest presentation of the band-folding effect in this system. We have revised the wording in the main text [near Eq.(2)] and the supplementary information (S4, bosonization) in an effort to better explain the band-folding effect.

1st Referee: *Second, and perhaps most importantly, can the effect of Umklapp scattering between spinons be seen in experiments that probes their transport or other thermal properties? One of the amazing thing about a critical system like here is that the heat-capacity has a residual gamma-term, so I am wondering if the fingerprint of Umklapp processes can be see from thermal transport measurements on this sample? Even if experiments cannot be done, perhaps predictions on how transport and thermal properties of YbAlO_3 are expected to be impacted by this discovery of Umklapp processes would be quite informative and kind of « close the loop »*

We are grateful to the Referee for this valuable suggestion concerning the more general implications of multiple fermion scattering. Certainly the residual heat-capacity term reflects the large density of states of low-energy quasiparticles, both for fermions and for fermionic magnetic systems, and these also contribute to the thermal conductivity because $\kappa \propto C$ in a ballistic framework [PhysRevB.64.054412].

If we consider the hypothetical purely magnetic contribution to transport in YbAlO_3 , we might expect a spinon contribution (similar to SrCuO_2 and other clean spin- $1/2$ chain compounds) in the temperature range between $T_N \sim 0.8$ K and $T \sim J/k_B \sim 2.5$ K. At higher T , the magnetic quasiparticles will be damped by the temperature, while in the AFM phase the transport would be suppressed due to the spin gap induced by the commensurate staggered field.

The in-field SDW ordering also causes a small gap Δ at the Fermi level due to h_{ic} , and this should suppress the spinon conductance at low temperatures, $k_B T < \Delta$, when compared to the single chain in an external field, which shows a truly gapless spectrum below saturation, $H < 2J$. However, in real magnetic systems, including YbAlO_3 , there are two major complications to observe this effect: (i) a possible spin-phonon interaction, which entangles lattice and magnetic contributions to the thermal transport (although an advantage of YbAlO_3 is that the optical phonons, which are primarily responsible for this interaction, are only weakly activated in the temperature range of interest). (ii) At first order in perturbation theory, the strength of the multiple scattering scales with (J_{ab}/J) ,

which makes it only a minor addition to the total magnetic signal. Altogether, these issues make thermal transport experiments rather challenging, although the outcome might be very valuable.

We note further that the same effect can, in principle, be seen in specific-heat measurements. Here we would expect to observe activated behaviour, $C \propto \exp\{\Delta/k_B T\}$, at $T \ll \Delta$ within the SDW phase. The temperature range for this is very low in YbAlO_3 ($\Delta < 0.04 \text{ meV} \sim 0.4 \text{ K}$), according to our INS data, meaning that the specific heat should be measured at least down to 0.04 K. Such experiments are difficult in materials like YbAlO_3 , which are good insulators with very poor coupling to the thermal bath at low temperatures. Another difficulty is caused by the significant nuclear contributions of Yb nuclei, which dominate the magnetic specific-heat signal in Yb-based systems below approximately 0.3 K (A. Steppke *et al.*, Phys. Status Solidi B **247**, No. 3, 737-739 (2010)).

We have collaborators who are currently performing measurements of magnetic thermal conductivity in YbAlO_3 . Their work is intended to address the low-temperature thermal properties of YbAlO_3 in detail, including the possible contributions from multiple fermion scattering. Therefore, to avoid any interference with their ongoing research and the impact of their conclusions, we would like to add only one paragraph to the present paper, the new penultimate paragraph of the main text, where we summarize in broad terms the impact of multiple fermion scattering on the thermal transport and encourage further experimental investigation of this effect. We thank the Referee in advance for her or his understanding of this situation.

2nd Referee: *In this paper, the authors have found satellite Bragg peaks beside the main peak of the incommensurate order for the Heisenberg chains coupled with anisotropic inter-chain couplings, using the neutron diffraction experiment. Also, reliable theoretical analyzes including numerical simulations clarified that the satellite peaks are cooperatively induced by the magnetic field and the anisotropic inter-chain couplings. I think that this finding has clarified a novel aspect of the field induced orders for quantum spin systems.*

We thank the Referee for his or her thorough reading of the manuscript and for remarking on the novelty of the results we present.

2nd Referee: *In the present version of the manuscript, however, I think that some explanations based on the fermion picture, which are especially emphasized in this paper, sound slightly misleading. In particular, the band structure in Fig. 1(a) and Fig. 3(b) is that of the free fermion (XY chain), not that of the Heisenberg chain. Actually, the spin wave velocity of the Heisenberg chain is nonperturbatively renormalized to $\pi J/2$ from J due to the interaction ($S^z S^z$ -term), which leads the effective coupling of $u = \pi J$ in the bosonic field theory (2) in which the intra-chain couplings were already involved. Also, the*

inter-chain coupling of Eq. (S5) is based on the effective bosonic field of Eq. (2). I therefore think that although the theoretical and numerical calculations are correct, the free-fermion-based explanation in this paper should be properly modified.

We thank the referee for this comment. Indeed we did not intend to give the impression that our results are all explained by a free-fermion picture. To understand the satellite peaks observed in neutron diffraction measurements, we study the low-energy properties of the spin model for YbAlO₃. The model consists of Heisenberg chains with Ising interchain couplings. The low-energy physics of a Heisenberg chain in a finite magnetic field can be understood, after a spin-to-fermion mapping, in terms of left- and right-moving fermions. One can show via the bosonization that for any finite field the fermion interaction (intrachain S^zS^z term) is irrelevant at low energies, and thus that the system is described by the Luttinger model given in Eq. (2) of the main text. As correctly pointed out by the referee, the effective coupling of this model is strongly renormalized by the interactions. The model for YbAlO₃ is unique because the anisotropic Ising interchain interaction is a relevant perturbation, which activates the Umklapp process between the left- and right-moving fermions and stabilizes the SDW order. We use bosonization to show formally the irrelevance of the intrachain interaction [in Eq. (S4)] and the relevance of the interchain interaction [in Eq. (S5)]. However, we find that it is more transparent to interpret the bosonization results by mapping back to the fermions (and the 1st Referee did request simple explanations). In the fermionic representation, the scattering due to the interchain interaction gives rise to the band-folding effect shown in Fig. 3(b). We note that the fermionic description (and correspondingly the bosonization result) is valid at low energies, where the bands can be linearized. In this picture, the left- and right-moving fermions describe collective modes near $\pm\pi$, and hence are different from free fermions. We agree with the referee that Figs. 1(a) and Fig. 3(b) in the previous version were quantitatively inaccurate as regards the y-axis scale and thank him or her for raising this point. In the updated version, we have modified this aspect and have also rephrased the related discussions in the text to make the fermion representation clearer.

2nd Referee: *Also, the inter-chain interaction effect on the dynamical spin structure factors of 1D quantum spin chains are well described by the Zeeman ladder approach:*

Prog. Theor. Phys. 63, 743 (1980)

Phys. Rev. Lett. 114, 017201 (2015)

I think that these papers should be included in the references. Then, the authors should describe how the satellite Bragg peaks appear in AbAlO₃ beyond the Zeeman ladder approach.

We thank the referee for mentioning these references. However, the Zeeman-ladder approach is not directly relevant to the observed satellite peaks in YbAlO_3 . The satellite peaks are observed in the incommensurate SDW phase for fields $H > 0.35$ T, while the Zeeman ladders appear in the commensurate Ising AFM phase (at $H < 0.35$ T in YbAlO_3), and there is a first-order transition between the two phases. Although both phases are associated with the interchain interactions, their underlying physics is very different. As explained both in the manuscript and in this reply, the satellite peaks in YbAlO_3 are caused by the multiple Umklapp process induced by the interchain interaction. By contrast, the Zeeman ladders reflect the spinon confinement caused by these couplings. Concerning the references mentioned by the referee, the 1980 paper discusses the dynamical correlation function of a single Heisenberg-Ising chain, and does not consider the effects of interchain interaction, as a result of which it is not relevant to our study. The 2015 paper discusses the Zeeman ladder at zero applied field, which is caused by the interchain coupling arising in the structurally complex Heisenberg-Ising compound $\text{BaCo}_2\text{V}_2\text{O}_8$. We have cited this work [Ref. 29] and added a short discussion in the revised manuscript.

2nd Referee: *In a QMC simulation for a finite chain length L , the wavenumber in the chain direction is generally quantized to be $2\pi/L \times (\text{integer})$, which may not be compatible with the incommensurate wavenumber associated with the magnetic field. I hope that the authors provide some comment about this finite size effect for $S(q,w)$.*

We are grateful to the referee for raising this interesting point. In principle, the incommensurate wave number q should not be compatible with $(2\pi/L)n$ for any finite system size L , as observed by the referee. However, in the simulation of a finite system with length L , a certain integer number n is found such that $(2\pi/L)n$ is a good approximation to q , i.e. n satisfies $(2\pi/L)n < q < (2\pi/L)(n+1)$. The error of the wavevector in the simulation is then less than $2\pi/L$, and the approximation improves for larger L . Our QMC data were obtained using a system with $L = 128$, where the finite-size effects on $S(q)$ are very small (as one may observe in Fig. S7 of the supplementary information). Following the suggestion of the referee, in the revised manuscript we have added further discussion of these finite-size effects in the supplementary information.

Reviewers' comments:

Reviewer #1 (Remarks to the Author):

This manuscript was much improved and I support publication in Nature Communications.

Reviewer #2 (Remarks to the Author):

2nd report of NCOMMS-20-30042A

In the new version of manuscript, the previous naïve explanation based on the free fermion picture was basically revised and the fermionic bands in Fig. 1 and Fig. 3(b) can be interpreted as an effective fermion renormalized by the intrachain interaction. However, I think that the main conclusion of this paper, i.e. confirmation of the fermion band structure and Umklapp scattering sounds overemphasized from theoretical viewpoint. There are two reasons. The first one is that the effective fermion is highly renormalized by the interaction effect. This implies that if a number of effective fermions is changed, the band structure of effective fermions is also renormalized. Exactly speaking, thus, a naive application of the free fermion band picture is not appropriate for such a highly interacting spin system as Heisenberg spin chain (Actually, the spin wave velocity and the Luttinger parameter have a magnetic-field dependence). Another reason is that, in the context of the interchain mean-field theory used in this paper, the soft mode in a one-dimensional spin liquid is generally stabilized by (unfrustrating) interchain couplings to form the corresponding three-dimensional long range order. For instance, the incommensurate order in BaCo₂V₂O₈ and SrCo₂V₂O₈ can be also interpreted as a result of the Umklapp scattering due to the interchain interaction. A particular point about YbAlO₃ is that its interchain couplings have the Ising-like anisotropy, while the intrachain chain coupling is isotropic, implying that the interchain couplings may induce both of the effective Ising anisotropy in the one-dimensional chain and the Umklapp scattering. This is an interesting point of this compound enabling an experimental observation of the multiple scattering peaks, but it does not affect a qualitative mechanism of the incommensurate order. As commented by the first referee, I also think that the scope of this paper seems to be rather limited, compared with the criterion of Nature communication.

As mentioned in the previous report, however, I think that the experimental and numerical data in the paper are solid, providing the direct evidence of multiple scattering of spinons due to interchain couplings. Thus I would like to support that this paper should be published in a more specific journal.

We very much appreciate the time and effort that both Reviewers spent on our work and we very much agree that the questions raised by Reviewer 2 are important: this is why we addressed them already, in the revised version of the manuscript and the detailed reply to both Reviewers that we resubmitted after the first round. Thus we can only conclude that these changes and discussion paragraphs have been unfortunately missed by Reviewer 2, and as a result we also find that the criticisms of this Reviewer are based on misunderstanding both our theoretical approach and the experimental situation. We describe these misunderstandings by separating them into 3 points.

(1) Reviewer 2 points out that a naïve free-fermion band picture is not appropriate to describe magnetic behaviour in a Heisenberg spin chain. This is a correct statement. However, we did not ever state that we use a simple free-fermion approximation, and this is why we used a bosonization approach. Within this picture, it is well known that the system can be described by a renormalized Luttinger model, and indeed the Fermi velocity is renormalized from the XY limit. Crucially for our current purposes, the multiple scattering processes occur at $E \rightarrow E_F$ and thus are robust under renormalization whenever $v_F > 0$. We stated the limits to the applicability of our approach several times throughout the manuscript. Further, we note that this mapping of Heisenberg (and even Ising-like XXZ) spin chains, although valid only in the low-energy sector, is used widely to describe the magnetic behaviour of spin-chain materials, and one recent example was published in Nature Communications [*W. J. Gannon et al., Nature Commun. 10, 1123 (2019)*]. Thus we are forced to disagree: it is not true that describing the **low-energy** behavior in terms of free fermions is invalid.

Reviewer 2 noted correctly that the spin-wave velocity and the Luttinger parameter should vary with the magnetic field, and so does the Fermi velocity. This is indeed one outcome of our theory. In this context, we realized belatedly that the sketches in Figs. 1(a) and 3(b) of our revised manuscript were somewhat inaccurate: instead of a rigid shift of the chemical potential, both the fermion dispersion and k_F should have been shown clearly to vary with the magnetic field. We have now revised these figures to highlight this aspect of our theory and we include them in the attached version of the manuscript. We trust that this will assist the Reviewer and reader in understanding the full situation.

(2) Reviewer 2 refers to the well-known one-dimensional compounds BCVO and SCVO, which exhibit SDW phases not dissimilar to the one observed in YbAlO₃. She or he claims that BCVO and SCVO develop three-dimensional order due to Ising-like anisotropy of interchain interaction and that the qualitative mechanism for the formation of SDW is the same in all cases. This is untrue. In BCVO and SCVO, the SDW instability comes from the **intrachain** Ising anisotropy. By contrast, in YbAlO₃ the SDW order is induced by the **interchain** Ising anisotropy alone, given that the intrachain coupling is of Heisenberg type. Nevertheless, despite these completely different mechanisms for SDW order, the multiple scattering process occurring via the interchain couplings exists in all of these compounds. This happens because the multiple scattering process becomes possible once

a SDW order is stabilized, regardless of the origin of this SDW order. The existence of the multiple scattering process in all compounds is therefore a strong statement that this process is quite generic in quasi-1D spin systems, contradicting the Reviewer's statement that our results are of "limited scope."

(3) We would like to stress that, while the observation and theoretical explanation of SDW ordering in YbAlO_3 is crucial, it is not the primary outcome of our work. Our key message is the experimental observation of the satellites at integer multiples of the primary incommensurate wavevector, $\mathbf{q} = 2n\mathbf{k}_F$, and our comprehensive theoretical modelling of these satellites. It is the latter step which associates the observed satellites directly with multiple scattering of fermions in YbAlO_3 . This is a new phenomenon, which to the best of our knowledge has not been observed before, either in low-dimensional metals or in one-dimensional magnets (including BCVO and SCVO). Thus, our results provide valuable new insight into the physics of correlated electron system.

We thank the Reviewer very much in advance for considering each of these points.

Sincerely,

Stanislav Nikitin,
Rong Yu
for all the authors

REVIEWERS' COMMENTS

Reviewer #2 (Remarks to the Author):

3rd report of NCOMMS-20-30042A

The questionable points in my previous report are basically fixed with this revision. As commented in the previous report, the quality of experiments and numerical computations is basically good. I therefore think that the present version of manuscript is acceptable for Nat. comm.